# *Staphylococcus aureus* and CA-MRSA Carriage among Brazilian Indians Living in Peri-Urban Areas and Remote Communities

**DOI:** 10.3390/antibiotics12050862

**Published:** 2023-05-06

**Authors:** Lígia Maria Abraão, Carlos Magno Castelo Branco Fortaleza, Carlos Henrique Camargo, Thaís Alves Barbosa, Eliane Patrícia Lino Pereira-Franchi, Danilo Flávio Moraes Riboli, Luiza Hubinger, Mariana Fávero Bonesso, Rodrigo Medeiros de Souza, Maria de Lourdes Ribeiro de Souza da Cunha

**Affiliations:** 1Department of Infectology, Dermatology, Diagnostic Imaging and Radiotherapy, Medical School (FMB) of Sao Paulo State University (UNESP), Botucatu 18618-970, Brazil; carlos.fortaleza@unesp.br (C.M.C.B.F.);; 2Nursing Research and Care Practices, Hospital Samaritano Higienopolis, São Paulo 01232-010, Brazil; 3Center of Bacteriology, Adolfo Lutz Institute—IAL, São Paulo 01246-000, Brazil; 4Department of Chemical and Biological Sciences, Biosciences Institute, UNESP—Universidade Estadual Paulista, Botucatu 18618-691, Brazil; 5Department of Nursing, Federal University of Acre—UFAC, Cruzeiro do Sul 69920-900, Brazil

**Keywords:** *Staphylococcus aureus*, CA-MRSA, colonization, brazilian indians, ethnicity, remote communities

## Abstract

The emergence of Community-associated methicillin-resistant *Staphylococcus aureus* (CA-MRSA) infections among indigenous populations has been reported. Usually, indigenous communities live in extreme poverty and are at risk of acquiring infections. In Brazil, healthcare inequality is observed in this population. To date, there are no reports of CA-MRSA infections, and no active search for asymptomatic *S. aureus* carriage has been conducted among Brazilian Indians. The aim of this study was to investigate the prevalence of colonization with *S. aureus* and CA-MRSA among Brazilian Indians. We screened 400 Indians (from near urban areas and remote hamlets) for *S. aureus* and CA-MRSA colonization. The isolates were submitted to clonal profiling by pulsed-field gel electrophoresis (PFGE), and selected isolates were submitted to multilocus sequence typing (MLST). Among 931 specimens (nasal and oral) from different indigenous individuals in remote hamlets, *S. aureus* was cultured in 190 (47.6%). Furthermore, CA-MRSA was found in three isolates (0.7%), all SCC*mec* type IV. PFGE analysis identified 21 clusters among the *S. aureus* isolates, and MLST analysis showed a predominance of sequence type 5 among these isolates. Our study revealed a higher prevalence of *S. aureus* carriage among Shanenawa ethnicity individuals (41.1%). Therefore, ethnicity appears to be associated with the prevalence of *S. aureus* in these populations.

## 1. Introduction

Infections caused by methicillin-resistant *Staphylococcus aureus* (MRSA) are of particular concern because they are harder to treat due to resistance to some antibiotics and because they are no longer limited to healthcare settings [1,2]. On the other hand, globally, strains of methicillin-sensitive *S. aureus* (MSSA) have been associated with a considerable burden of invasive disease, especially among indigenous communities [2,3,4,5]. Colonization by *S. aureus* represents a risk factor for the occurrence of autogenous infections and cross-transmission to other individuals [6]. Historically, indigenous populations have been affected by high rates of infectious diseases [2,7,8]. Outbreaks of Community-associated methicillin-resistant *Staphylococcus aureus* (CA-MRSA) were first described among indigenous communities. A remarkable observation was made in Australia in 1980 when a strain of MRSA arose spontaneously in remote Aboriginal communities [9,10]. There is a global trend of considerably higher rates of invasive *S. aureus* disease among indigenous populations [11,12,13,14].

According to the most recent Brazilian Census [15], 305 indigenous ethnic groups are found across the Brazilian territory, with a population of 896,917 indigenous people living in the country: 324,834 in urban areas and 572,083 in rural hamlets. Ethnic groups and tribes vary widely in their customs and living conditions, including food, personal hygiene, religious rituals, and housing.

To date, there are no reports of CA-MRSA infections among Brazilian Indigenous, and no active search for asymptomatic *S. aureus* carriage and virulence profile has been conducted. To improve our understanding of *S. aureus* epidemiology among Brazilian Indians, we investigated the contribution of individual, household, and pathogen-related factors associated with *S. aureus* carriage. The objectives of the study were to investigate the prevalence of colonization with overall *S. aureus* and CA-MRSA and virulence factors among Brazilian Indians. We were especially interested in comparing populations living close to urban centers with those from remote areas.

## 2. Results

A total of 190 *S. aureus* isolates were recovered from the nasal and oral mucosa of 400 individuals (116 from the São Paulo [SP] state and 284 from the Acre [AC] state). The overall prevalence of *S. aureus* colonization was 47.6% (95% confidence interval [CI], 42.6–52.6%) and did not differ among the two study groups (49.5% in AC, 43.1% in SP). Only three subjects (all from AC) carried CA-MRSA, with a prevalence of 0.7% (95% CI: 0.19–2.37). In addition, the CA-MRSA isolates were tested for susceptibility to penicillin, ceftaroline, quinupristin, sulfamethoxazole, clindamycin, erythromycin, and levofloxacin, showing resistance only to penicillin. All CA-MRSA strains harbored SCC*mec* type IV.

Among all *S. aureus* strains, the prevalence of the gene coding for toxic shock syndrome toxin 1 (TSST-1; *tst*) was 6.5%. Among genes encoding enterotoxins, *sec* was the most prevalent (19.9%), followed by *seb* (14.1%) and *sea* (9.4%). The genes for exfoliative toxins A (*eta*) and B (*etb*) were found in 3.6% and 6.8% of the isolates, respectively. Furthermore, 96.3%, 20.4%, and 80.6% of the isolates harbored genes for hemolysins alpha (*hla*), beta (*hlb*), and delta (*hld*), respectively. Biofilm genes (*icaA*, *icaB*, *icaC*, and *icaD*) were detected in 82.2%, 1.0%, 7.8%, and 72.7% of the isolates, respectively. The gene coding for Panton–Valentine leukocidin (PVL; *lukS*-PV) was found in 36 (13.5%) isolates. It is worth noting that all of these isolates were methicillin-susceptible.

In the analysis of risk factors associated with the carriage of *S. aureus*, the univariate Poisson regression model revealed a positive association with age and the number of baths (Table 1). When the factor ethnicity was observed in which the Shanenawa group (the largest group among the indigenous populations studied) was used as a reference, there was a negative association between the Teregua and Kaxinawa ethnicities and the outcome (Table 2). The Shanenawa ethnicity showed a positive association with *S. aureus* carriage when analyzed as a dichotomous variable and was therefore used as the reference in relation to the other ethnicities.

However, in the multivariate analysis, ethnicity was the only independent factor associated with *S. aureus* carriage. The age variable showed a marginally significant *p*-value (*p* = 0.08). Although age and the number of baths did not remain in the multivariate model, they seem to contribute to the outcome studied.

Considering the power of the association of ethnicity with the outcome, univariate analyses were performed comparing the prevalence of virulence factors associated with the ethnic types in order to identify differences in pathogenicity between the *S. aureus* isolates that colonize the indigenous groups studied. From the same perspective, risk factors (habits and customs) were analyzed, and demographic variables (income and the number of household members) were associated with *S. aureus* carriage according to ethnic group. The results show that the Shanenawa ethnicity stands out in relation to the prevalence of the virulence genes, as well as the variables related to habits, customs, and demographics (Table 2, Table 3 and Table 4).

PFGE typing identified 21 clusters of S. aureus isolates, most of them grouping strains from SP and AC. The dendrogram in Figure 1 shows the electrophoresis typing of the main *S. aureus* clusters identified in indigenous populations from the southeast and northern regions of Brazil. Of these, the following 17 isolates were typed by MLST: ST 5 (4—the most prevalent), ST 8 (2), ST 25 (2), ST 97 (2), ST 188 (2), ST 1 (1), ST 6 (1), ST 15 (1), ST 1635 (1), and SLV 7067 (1).

## 3. Discussion

In this study, we found that ethnicity was the only predictor associated with *S. aureus* colonization in Brazilian Indians. Furthermore, we identified that income was associated with the prevalence of *S. aureus* carriage among almost all ethnicities. Previous studies also identified a possible association between race, ethnicity, and socioeconomic status, suggesting that these factors seem to contribute to the phenomenon of *S. aureus* colonization [15,16]. Ethnicity is related to the collectivity of individuals, and it is distinguished by their sociocultural specificity, mainly reflected in the language, religion, habits, and living conditions of populations. Considering these facts, it is important to emphasize that most Indian populations live in agglomerations under poor hygiene and sanitation conditions, with large socioeconomic disparities that can contribute to *S. aureus* carriage [17].

The traditional epidemiological factors for *S. aureus* colonization, such as young age, male sex, underlying comorbidities, smoking, and previous hospitalization, were not relevant in the population studied, in agreement with a study investigating an Aboriginal community in Canada [18].

Despite widespread knowledge of the process of *S. aureus* carriage as a predictor of *S. aureus* infections, in Brazil, little is known about the dynamics of colonization and infection with *S. aureus* among indigenous populations. Our study identified a prevalence of *S. aureus* and CA-MRSA carriage of 47.6% and 1.0%, respectively. CA-MRSA strains were restricted to indigenous people belonging to the Amazon region of Brazil, all of them harboring SCC*mec* IV. A study conducted on the Amerindian population of Wayampi in the village of Trois Sauts, an isolated region in the Amazon forest of French Guiana, demonstrated rates of nasal carriage and persistent colonization with *S. aureus* of 57.8% and 26%, respectively. None of the isolates showed resistance to methicillin [19].

Our findings revealed that CA-MRSA was rare and, as also reported in a population-based survey conducted in Brazil, all CA-MRSA harbored SCC*mec* IV [20]. On the other hand, the prevalence of *S. aureus* was higher than that reported for the general population [20]. Antimicrobial susceptibility testing of the MRSA isolates using disks impregnated with penicillin, ceftaroline, quinupristin, sulfamethoxazole, clindamycin, erythromycin, and levofloxacin revealed sensitivity to all antimicrobials, except for penicillin. All isolates were also resistant to penicillin. According to Sader et al. [21], sensitivity rates of CA-MRSA strains are generally higher when compared to hospital-associated MRSA, especially for clindamycin and levofloxacin. The same was observed for ceftaroline, a drug that has recently been introduced on the market as an option for the treatment of infections caused by MRSA.

The results obtained regarding the detection of virulence genes show diversity in the pathogenicity profile of *S. aureus* isolated from indigenous populations, similar to those already identified in non-indigenous populations [22]. Among the most prevalent virulence factors were genes coding for hemolysins *hla* and *hld* and the *icaA* and *icaD* genes involved in biofilm formation. According to Bride et al. [23], these genes are often produced by a large number of *S. aureus* strains. The findings obtained in the present study showed 13.5% of isolates carrying the PVL gene, none of them associated with CA-MRSA. According to Boan et al. [24], infections with PVL-positive *S. aureus* strains disproportionately affect young indigenous people or subjects with fewer healthcare-related risk factors. In a previous study involving a non-indigenous population, only five isolates were found to harbor the gene coding for PVL, corresponding to 2.2% of all isolates [22]. Considering the specific living conditions of the indigenous populations mentioned above, it is possible to infer that such factors may contribute to *S. aureus* carriage and the occurrence of skin infections associated with the PVL gene (*lukS*-PV) [24].

The univariate analyses compared the prevalence of virulence genes associated with ethnicity type and also analyzed the risk factors (habits, customs, and demographic factors) associated with *S. aureus* carriage according to ethnicity; significant results were obtained for the *sec*, *hld*, *tst*, *icaA*, and *icaD* genes. Regarding habits, customs, and demographics, all variables tested were statistically significant: skin pigmentation, number of baths, use of medicinal herbs, baths in the river, monthly income, and number of household members. It is worth noting that individuals of the Shanenawa ethnicity, which was used as a reference and showed a positive association with *S. aureus* when tested as a dichotomous variable, carried the most virulent and resistant *S. aureus* strains. Commonly, this ethnicity more frequently exhibits habits, customs, and demographic factors that may be associated with a higher prevalence of colonization by *S. aureus*.

Regarding PFGE analysis, 21 clusters circulating among the indigenous populations from southeastern and northern Brazil were identified. Of these, five major clusters (1, 4, 6, 10, and 14) contained a larger number of isolates that were more prevalent among the regions studied. Interesting data were observed for these clusters in which isolates from indigenous communities in the southeast formed clusters with isolates identified in the Amazon region. Another interesting finding was that, although not one of the most prevalent, cluster 7 contained sensitive isolates that grouped with one resistant isolate belonging to ST 5. According to Robinson and Enright [25], the dissemination of genotypes occurs not only among individuals but also through the ability to transmit the mobile genetic element SCC*mec* IV through MSSA strains. The remaining CA-MRSA isolates formed clusters with clone USA800. This clone represents a lineage of pandemic MRSA such as CC5 [26].

The sequence type (ST) number 5 was the most frequent. This ST is one of the most prevalent clonal complexes in both hospital and community settings [27,28]. Moreover, the other STs identified in the indigenous populations studied resemble those commonly disseminated in other types of populations [29,30]. ST 5 associates with different types of SCC*mec* (I, II, and IV) and represents the ancestor of the clone described as a Cordobes/Chilean clone and of the MRSA pediatric epidemic clone detected in 1999 [26,31].

## 4. Materials and Methods

### 4.1. Study Design, Subjects, and Procedures

A cross-sectional, population-based study was conducted. Individuals of indigenous populations from two different Brazilian states were included. One state was São Paulo, the largest urban center of the country with an area of 1,521,110 km^2^ and 11,253,503 inhabitants, and the other state was Acre, located in the remote region of the Amazon forest, which comprises a large area of 8,835,520 km^2^ and has 377,057 inhabitants according to IBGE [14].

Our sample size was established by calculating proportions, assuming a level of significance of 5% and accuracy of 5%, according to the number of individuals in each village/hamlet. A total of 400 subjects were included in the study. Of those, 116 lived in a peri-urban village in the city of Bauru, São Paulo State (SP), southeastern Brazil, and 284 lived in small hamlets in Acre State (AC), Brazilian Amazon region. Nasal and oropharyngeal swabs were collected from September to November 2014. Data on demographics (including ethnicity), habits and customs, comorbidities, and clinical variables were collected in interviews with the subjects. Table 5 lists the variables analyzed.

### 4.2. Specimen Collection, Culture, and Antimicrobial Susceptibility Tests

Nasopharyngeal and oral swabs were collected, transported in Stuart medium, and cultured on a selective medium (Baird-Parker Agar) for up to 48 h. *Staphylococcus aureus* was identified based on colony morphology and standard biochemical tests [32]. Susceptibility to methicillin was tested by disk diffusion using cefoxitin (30 µg) disks according to the recommendations of the Clinical Laboratory Standards Institute [33]. For the MRSA isolates, susceptibility to penicillin, clindamycin, levofloxacin, erythromycin, sulfamethoxazole–trimethoprim, quinupristin–dalfopristin, and ceftaroline was tested by disk diffusion and to oxacillin and vancomycin by the E test (BioMerieux) [32].

### 4.3. Identification of Virulence Genes

PCR assays for the detection of virulence genes (*sea*, *seb*, *sec*, *hla*, *hlb*, *hld*, *eta*, *etb*, *etd*, *icaA*, *icaB*, *icaC*, and *icaD*) were performed following established protocols [34,35,36,37,38,39]. The primers and references used in this step are listed in Table 6.

### 4.4. Molecular Methods

Methicillin resistance was also assessed through the detection of the *mec*A gene by real-time PCR performed in a LightCycler system (Roche) [39]. The staphylococcal cassette chromosome (SCC*mec*) was characterized using the protocol described by Milheiriço et al. [40]. Pulsed-field gel electrophoresis (PFGE) was performed on *Sma*I-digested chromosomal DNA according to the protocol described by McDougal et al. [41]. Band patterns were analyzed with BioNumerics 7.6 (Applied Maths), and a dendrogram was generated using the unweighted pair group method with arithmetic mean (UPGMA). Clusters were defined as any group of more than four isolates with a Dice similarity coefficient ≥ 80%, assuming tolerance and optimization of 0.5 and 1.25%, respectively [41].

Representative isolates of the PFGE clusters were submitted to multilocus sequence typing (MLST) [42]. The *arcC*, *aroE*, *glpF*, *gmk*, *pta*, *tpi*, and *yqIL* genes were amplified separately with the specific primers described by Enright et al. [26], and both strands were sequenced. Sequence types (ST) were assigned using the BioNumerics 7.6 software [26].

### 4.5. Epidemiological Analysis

Data were collected and analyzed using Epi Info for Windows, version 3.5.1 (© Centers for Disease Control and Prevention, Atlanta, GA, USA) and SPSS 20.0 (IBM, Armonk, NY, USA). The base of the study was formed by subjects (indigenous people) with a culture positive for *S. aureus*. Univariate and multivariate Poisson regression models were built. A stepwise forward strategy was used to select variables for the multivariate models, with a *p*-value of 0.05 as a limit for inclusion/removal [42,43].

Figure A1 (Appendix A) shows the workflow of the methodological steps.

### 4.6. Ethical Issues

This study was approved by the National Committee of Ethics in Research (CONEP/CNS/MS), CAAE: 08428912.3.0000.5411, Approval number 674.368.

## 5. Conclusions

In conclusion, ethnicity appears to be associated with a higher prevalence of *S. aureus* and virulence in special populations, even though the prevalence of CA-MRSA was low. These results might be related to specific habits and customs since poor housing conditions, hygiene, and sanitation are features present in most ethnical groups and can influence the carriage and dissemination of *S. aureus* among populations. It is, therefore, extremely important that these factors be considered for the development and implementation of strategies designed to control the spread of microorganisms among different human populations, especially considering the health inequality of indigenous populations in Brazil.

## Figures and Tables

**Figure 1 antibiotics-12-00862-f001:**
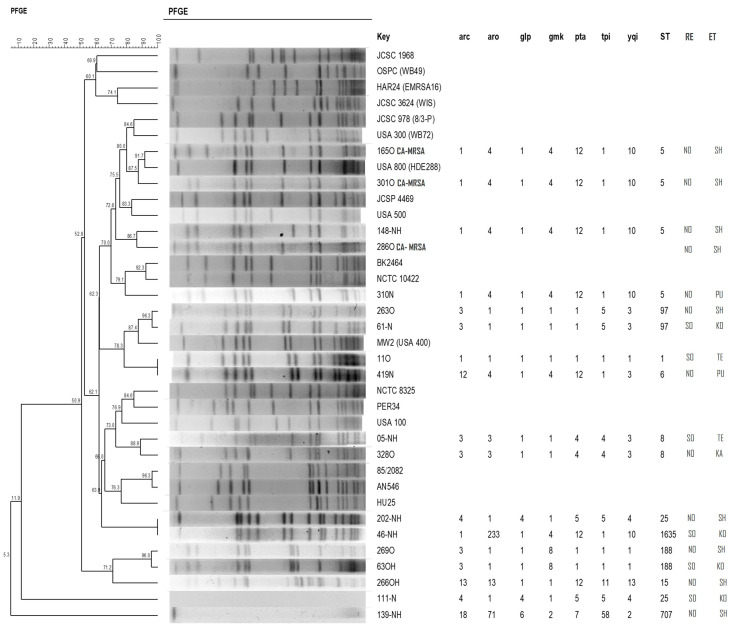
Dendrogram showing pulsed-field gel electrophoresis and multilocus sequence typing of the main S. aureus and CA-MRSA clusters identified in Indian populations from the southeast (SO) and north (NO) regions of Brazil. ST—sequence type; RE—region; ET—ethnicity; SH—Shanenawa; PU—Puyanawa; KO—Kopenoti; TE—Teregua; KA—Kaxinawa.

**Table 1 antibiotics-12-00862-t001:** Poisson regression model for the analysis of risk factors associated with *S. aureus*.

Univariate Analysis					Multivariate Analysis	
Predictors	*S. aureus*	Negative	(95% CI)	*p* Value	RR (95% CI)	*p* Value
Demographic Variables						
Women	113 (59.5)	112 (53.6)	1.12 (0.93–1.35)	0.24		
Age, median (quartiles)	21 (11.5–34)	27 (14–43)	…	0.03 *	0.99 (0.98–1.00)	0.08
Ethnicity						
*Shanenawa (reference) **	78 (41.1)	57 (27.3)	…	…	1.36 (CI, 1.02–1.82)	0.03 *
*Puyanawas I*	22 (11.1)	30 (14.4)	0.73 (0.52–1.04)	0.06	0,74 (0.46–1.20)	0.22
*Kaxinawa*	16 (8.4)	34 (16.3)	0.55 (0.36–0.85)	0.002 *	0.55 (0.32–0.94)	0.03 *
*Kopenoti*	31 (16.3)	33 (15.8)	0.84 (0.63–1.12)	0.22	0.87 (0.57–1.32)	0.52
*Teregua*	19 (10.0)	33 (15.8)	0.63 (0.43–0.93)	0.009 *	0.67 (0.40–1.11)	0.11
*Ashaninka*	4 (2.1)	1 (10.5)	1.34 (0.87–2.20)	0.32	1.32 (0.48–3.62)	0.58
*Puyanawas II*	21 (11.1)	21 (10.1)	0.87 (0.62–1.21)	0.38	0.91 (0.56–1.38)	0.7
Group (São Paulo x Acre)	50 (26.3)	66 (31.6)	0.89 (0.73–1.08)	0.25		
Income in R$, median (quartiles)	688 (430–800)	700 (422–815)		0.97		
Schooling						
*Illiterate (reference)*	13 (6.8)	23 (11.0)	…	…		
*Incomplete elementary school*	77 (40.5)	91 (43.5)	1.27 (0.80–2.002)	0.29		
Continuing						
Univariate Analysis	Multivariate Analysis	Univariate Analysis	Multivariate Analysis	Univariate Analysis	Multivariate Analysis	Univariate Analysis
*Complemente elementary school*	56 (29.5)	46 (22.0)	1.52 (0.45–2.43)	0.052		
*Complete high school*	39 (20.5)	41(19.6)	1.35 (0.83–2.20)	0.21		
*College degree*	5 (2.6)	8 (3.8)	1.07 (0.47–2.40)	1.0		
Brickhouse	60 (31.6)	71 (33.9)	0.95 (0.78–1.16)	0.61		
Sewerage system	85 (44.7)	83 (39.7)	1.10 (0.91–1.51)	0.31		
Number of household members, median (quartiles)	5.5 (4.5–7)	5.0 (3.0–7.5)				
Distance from the health unit, median (quartiles)	2.5 (2.5–5.0)	2.0 (2.0–5.5)		0.52		
Habits and Customs						
Tattoo Collective sports	18 (9.4) 87 (45.7)	24 (11.4) 93 (44.5)	0.90 (0.68–1.20) 1.02 (0.84–1.23)	0.62 0.80		
Earrings	45 (32.3)	23 (44.2)	0.69 (0.44–1.10)	0.17		
Skin pigmentation	70 (36.8)	76 (36.3)	1.00 (0.83–1.22)	0.92		
Urucum	55 (29.0)	68 (32.5)	0.92 (0.75–1.12)	0.50		
Jenipapo	67 (35.2)	38 (59.3)	0.85 (0.68–1.08)	0.22		
Number of daily baths, median (quartiles)	3.5 (2.5–4.0)	3.0 (2.0–4.0)	…	0.04 *		
Baths in the river	61 (43.7)	79 (37.8)	0.88 (0.73–1.07)	0.27		
Showers	131 (68.9)	135 (64.5)	1.09 (0.90–1.32)	0.35		
Use of medicinal herbs	33 (17.)	40 (19.1)	0.94 (0.74–1.19)	0.64		
Use of herbal drink for rituals	1 (0.53)	6 (2.87)	0.60 (0.44–0.82)	0.07		
Use of topical medicinal substances	26 (40.6)	38 (59.3)	0.85 (0.68–1.08)	0.22		
Drinking	29 (15.2)	32 (15.3)	0.99 (0.77–1.29)	0.98		
Smoking	34 (17.8)	42 (20.1)	0.93 (0.74–1.17)	0.57		
Clinical Variables						
Diabetes	8 (4.2)	11 (5.2)	0.90 (0.60–1.33)	0.62		
Skin infections A	3 (1.58)	2 (0.96)	1.31 (0.44–3.85)	0.57		
Skin infections R	6 (3.16)	8 (3.83)	0.91 (0.57–1.45)	0.71		
Antibiotics	22 (11.5)	19 (9.0)	1.14 (0.81–1.61)	0.41		
Recent outpatient consultations	32 (16.8)	26 (12.4)	1.19 (0.88–1.61)	0.21		
Surgery	17 (8.95)	11 (5.26)	1.35 (0.84–2.17)	0.15		
Hospitalization	27 (14.2)	34 (16.2)	0.92 (0.72–1.18)	0.56		
Pneumonia	1 (0.53)	1 (0.48)	1.04 (0.26–4.20)	0.94		

Data in numbers (%). RR, rate ratio (prevalence ratio); CI confidence interval. * Statistically significant data.

**Table 2 antibiotics-12-00862-t002:** Results of the univariate analysis of risk factors associated with *S. aureus* carriage according to ethnicity.

	Shanenawa	Puyanawas I	Puyanawas II	Kaxinawa	Kopenoti	Teregua
Age	21 (11–33)	27 (11–38)	27 (11–49) *	20 (10–34) *	27 (13–42)	31 (13–54) *
Women	71 (52.6%)	24 (47.1%)	29 (69.0%)	28 (56%)	40 (62.5%)	30 (57.7%)
*S. aureus*	78 (57.8%)	21 (41.2%)	21 (50.0%)	16 (32.0%) *	31 (48.4%)	19 (36.5%)
MRSA	3 (2.2%)	0 (0.0%)	0 (0.0%)	0 (0.0%)	0 (0.0%)	0 (0.0%)
Number of household members, median (quartiles)	6 (4–8)	5 (4–5) *	4 (3–8) *	6 (4–8) *	4 (4–6) *	4 (3–6) *
Number of baths, median (quartiles)	3 (3–4)	3 (3–4)	4 (3–4)	3 (3–4)	2 (1–2) *	2 (2–3) *
Baths in the river	68 (50.4%)	15 (29.4%)	9 (21.4%) *	43 (86.0%) *	0 (0.0%) *	0 (0.0%) *
Tattoo	31 (23%)	3 (5.9%) *	1 (2.4%) *	4 (8.0%) *	0 (0.0%) *	3 (5.8%) *
Skin pigmentation	76 (56.3%)	22 (43.1%)	17 (40.5%)	26 (52.0%)	0 (0.0%) *	0 (0.0%) *
Earrings	1 (0.7%)	0 (0.0%)	0 (0.0%)	0 (0.0%)	1 (1.6%)	2 (3.8%)
Smoking	40 (29.6%)	4 (7.8%) *	7 (16.7%)	10 (20.0%)	7 (10.9%) *	7 (13.5%)
Use of snuff for rituals	1 (0.7%)	1 (2.0)	0 (0.0%)	0 (0.0%)	0 (0.0%)	0 (0.0%)
Use of herbal drink for rituals	1 (0.7%)	6 (11.8%) *	0 (0.0%)	0 (0.0%)	0 (0.0%)	0 (0.0%)
Use of medicinal herbs **	39 (28.9%)	19 (37.3%)	5 (11.9%)	7 (14%)	0 (0.0%) *	1 (1.9%) *
Drinking	39 (28.9%)	5 (9.8%) *	3 (7.1%) *	9 (18.0%)	3 (4.7%) *	1 (1.9%) *
Collective sports	80 (59.3%)	31 (60.8%)	20 (47.6%)	33 (66%)	5 (7.8%) *	10 (19.2%) *
Brickhouse	2 (1.5%)	4 (7.8%)	8 (19.0%) *	3 (6.0%)	63 (98.4%) *	51 (98.1%) *
Income in R$, median (quartiles)	632 (334–800) *	600 (400–700) *	670 (490–1000) *	550 (200–684) *	700 (550–750) *	760 (700–1150)

Data in numbers (%).* Statistically significant data. ** Mixture of medicinal herbs typically used by indigenous people for the empirical treatment of diseases.

**Table 3 antibiotics-12-00862-t003:** Results of the univariate analysis of risk factors associated with *S. aureus* carriage according to ethnicity.

	Ethnicities—São Paulo & Acre														
	Shanenawa (%)		Puyanawa I (%)		Kaxinawa (%)		Kopenoti (%)		Teregua (%)		Ashaninka (%)		Puyanawas II (%)		*p* Value
Habits and Customs	Positive	Negative	Positive	Negative	Positive	Negative	Positive	Negative	Positive	Negative	Positive	Negative	Positive	Negative	
Skin pigmentation	76 (52.1)	59 (23.3)	22 (15.1)	29 (11.5)	26 (17.8)	24 (9.5)	-	64 25.6)	-	52 (20.6)	5 (3.4)	-	17 (11.6)	25 (9.9)	<0.01 *
Use of medicinal herbs **	39 (53.4)	96 (29.4)	19 (26.0)	32 (9.8)	7 (9.6)	43 (13.2)	-	64 (19.6)	1 (1.4)	51 (15.6)	2 (2.7)	3 (0.9)	5 (6.8)	37 (11.3)	<0.01 *
Baths in the river	68 (48.6)	67 (25.9)	15 (10.7)	36 (13.9)	43 (30.7)	7 (2.7)	-	64 (24.7)	-	52 (20.1)	5 (3.6)	-	9 (6.4)	33 (12.7)	<0.01 *
Number of baths, median (quartiles)	3.2 (1–4)		3.1 (1–4)		3.2 (1–4)		1.7 (1–4)		2.1 (1–4)		3.0 (2–4)		3.5 (2–4)		
Income in R$, median (quartiles)	696 (50–4000)		781 (70–3550)		505 (60–2000)		810 (134–4000)		1013 (300–2500)		710 (250–1400)		837 (649–4000)		<0.01 *
Household members, median (quartiles)	6.3 (1–20)		4.0 (1–9)		6.0 (2–11)		4.6 (1–9)		3.9 (1–6)		8.0 (8–8)		5.2 (2–11)		<0.01 *

Data in numbers (%). Positive: *S. aureus*. * Statistically significant data. ** Mixture of medicinal herbs typically used by indigenous people for empirical treatment of diseases.

**Table 4 antibiotics-12-00862-t004:** Results of the univariate analysis of virulence factors associated with ethnicities.

Univariate Analysis-Ethnicities—Acre & Sao Paulo State															
	Shanenawa (%)		Puyanawa I (%)		Kaxinawa (%)		Ashaninka (%)		Puyanawas II (%)		Kopenoti-Sp (%)		Teregua-Sp (%)		
Genes	Positive	Negative	Positive	Negative	Positive	Negative	Positive	Negative	Positive	Negative	Positive	Negative	Positive	Negative	*p* Value
*S. aureus* (*sau*)	78 (41.1)	57 (27.3)	21 (11.1)	30 (14.4)	16 (8.4)	34 (16.3)	4 (2.1)	1 (0.5)	21 (11.1)	21 (10.0)	31 (16.3)	33 (15.8)	19 (10.0)	33 (15.8)	0.01 *
*Sea*	7 (38.9)	72 (41.6)	1 (5.6)	20 (11.6)	-	16 (9.2)	1 (5.6)	3 (1.7)	1 (5.6)	20 (11.6)	6 (33.3)	25 (14.5)	2 (11.1)	17 (9.8)	0.27
*Seb*	10 (37.0)	69 (42.1)	1 (3.7)	20 (12.2)	3 (11.1)	13 (7.9)	-	4 (2.4)	2 (7.4)	19 (11.6)	7 (25.9)	24 (14.6)	4 (14.8)	15 (9.1)	0.47
*Sec*	18 (47.4)	61 (39.9)	5 (13.2)	16 (10.5)	3 (7.9)	13 (8.5)	3 (7.9)	1 (0.7)	4 (10.5)	17 (11.1)	5 (13.2)	26 (17.0)	-	19 (12.4)	0.03 *
*Hla*	75 (40.8)	4 (57.1)	20 (10.9)	1 (1.3)	15 (8.2)	1 (14.3)	4 (2.2)	-	20 (10.9)	1 (14.3)	31 (16.8)	-	19 (10.3)	-	0.81
*Hld*	66 (42.9)	13 (35.1)	13 (8.4)	8 (21.6)	11 (7.1)	5 (13.5)	3 (1.9)	1 (2.7)	14 (9.1)	7 (18.9)	29 (18.8)	2 (5.4)	18 (11.7)	1 (2.7)	0.02 *
*PVL*	14 (38.9)	65 (41.7)	3 (8.3)	18 (11.5)	2 (5.6)	14 (9.0)	1 (2.8)	3 (1.9)	5 (13.9)	16 (10.3)	9 (25.0)	22 (14.1)	2 (5.6)	18 (11.5)	0.62
*Eta*	5 (71.4)	74 (40.2)	-	21 (11.4)	1 (14.3)	15 (8.2)	-	4 (2.2)	-	21 (11.4)	1 (14.3)	30 (16.3)	-	19 (10.3)	0.62
*Etd*	5 (38.5)	74 (41.6)	-	21 (11.8)	2 (15.4)	14 (7.9)	-	4 (2.2)	-	21 (11.8)	6 (46.2)	25 (14.0)	-	19 (10.7)	0.03
*TSST-1*	1 (8.3)	75 (43.6)	-	21 (12.2)	-	15 (8.7)	-	3 (1.7)	-	20 (11.6)	4 (33.3)	26 (15.1)	7 (58.3)	12 (7.0)	<0.01 *
*icaA*	65 (41.4)	14 (41.2)	20 (12.7)	1 (2.9)	15 (9.6)	1 (2.9)	-	4 (11.8)	20 (12.7)	1 (2.9)	22 (14.0)	9 (26.5)	15 (9.6)	4 (11.8)	<0.01 *
*icaD*	53 (38.1)	26 (50.0)	17 (12.2)	4 (7.7)	13 (9.4)	3 (5.8)	1 (0.7)	3 (5.8)	20 (14.4)	1 (1.9)	21 (15.1)	10 (19.2)	14 (10.1)	10 (19.2)	004 *

Data in numbers (%). Positive: *S. aureus*. * Statistically significant data.

**Table 5 antibiotics-12-00862-t005:** Variables included in the analysis.

Variables Analyzed		
Category	Description	Examples or Additional Information
Demographic	Gender, age	
	Ethnicity	
	Income in R$	
	Schooling	Illiterate, incomplete elementary school, complete elementary school, complete high school, college degree
	Type of housing	Brickhouse (house built with bricks and cement), houses made of wood/straw or rammed earth
	Sewerage system	
	Median number of household members	
	Distance from the health unit in kilometers	
Habits and customs	Tattoo	
	Collective sports	
	Earrings	
	Skin pigmentation	In general, indigenous peoples have the habit of body painting. In Brazil, dyes are made from natural compounds, such as Urucum and Jenipapo. These dyes remain on the skin for a period of 15 to 20 days.
	Urucum	
	Jenipapo	
	Number of daily baths	Indigenous people living in remote areas have the habit of bathing in the river, and they commonly take many baths throughout the day.
	Baths in the river	
	Use of medicinal herbs	
	Use of herbal drink for rituals	Ayahuasca
	Use of topical medicinal substances	
	Drinking	
	Smoking	Snuff, cigarette
Clinical variables	Diabetes	
	Skin infections A (in the last year)	
	Skin infections R (recent)	
	Antibiotic use	
	Recent outpatient consultations	
	Surgery	
	Hospitalization	
	Pneumonia	

**Table 6 antibiotics-12-00862-t006:** Primers used for the amplification of toxin and biofilm genes.

Virulence Factor	Gene	Primer	Sequence (5′-3′)	Product	Control	References
Enterotoxin A	*Sea*	SEA-1	TTGGAAACGGTTAAAACGAA	120 bp	ATCC13565	[34,35]
		SEA-2	GAACCTTCCATCAAAAACA			
Enterotoxin B	*seb*	SEB-1	TCGCATCAAACTGACAAACG	478 bp	ATCC 14458	[34,35]
		SEB-2	GACGGTACTCTATAAGTGCC			
Enterotoxin C	*Sec*	SEC-1	GACATAAAAGCTAGGAATTT	257 bp	ATCC 19095	[34,35]
		SEC-2	AAATCGGATTAACATTATCC			
		SEE-2	TAACTTACCGTGGACCCTTC			
Toxic shock syndrome toxin 1	*Tst*	TSST-1	ATGGCAGCATCAGCTTGATA	350 bp	N315	[36]
		TSST-2	TTTCCAATAACCACCCGTTT			
Exfoliative toxin A	*Eta*	ETA-1	CTAGTGCATTTGTTATTCAA	119 bp	N5	[37]
		ETA-2	TGCATTGACACCATAGTACT			
Exfoliative toxin B	*Etb*	ETB-1	ACGGCTATATACATTCAATT	200 bp	ZM	[37]
		ETB-2	TCCATCGATAATATACCTAA			
Exfoliative toxin D	*Etd*	ETD-1	AACTATCATGTATCAAGG	376 bp		[37]
		ETD-2	CAGAATTTCCCGACTCAG			
Hemolysin α	*hla*	HLA-1	CTGATTACTATCCAAGAAATTCGATTG	209 bp	N315	[38]
		HLA-2	CTTTCCAGCCTACTTTTTTATCAGT			
Hemolysin β	*hlb*	HLB-1	GTGCACTTACTGACAATAGTGC	309 bp	RN4420	[38]
		HLB-2	GTTGATGAGTAGCTACCTTCAGT			
Hemolysin δ	*hld*	HLD-1	ATGGCAGCAGATATCATTTC	357 bp	N315	[38]
		HLD-2	CGTGAGCTTGGGAGAGAC			
Biofilm	*ica*A	*ica*A-1	ACA GTC GCT ACG AAA AGA AA	103 bp		[39]
		*ica*A-2	GGA AAT GCC ATA ATG AGA AC			
Biofilm	*ica*B	*ica*B-1	CTG ATC AAG AAT TTA AAT CAC AAA	302 bp		[39]
		*ica*B-2	AAA GTC CCA TAA GCC TGT TT			
Biofilm	*ica*C	*ica*C-1	TAA CTT TAG GCG CAT ATG TTT	400 bp		[39]
		*ica*C-2	TTC CAG TTA GGC TGG TAT TG			
Biofilm	*ica*D	*ica*D-1	ATG GTC AAG CCC AGA CAG AG	198 bp		[39]
		*ica*D-2	CGT GTT TTC AAC ATT TAA TGC AA			

## Data Availability

The data presented in this study are original and have not been published in scientific journals. The only document that contains these data is the doctoral thesis of Lígia Maria Abraão, openly available in [Institutional Repository of UNESP] at [https://repositorio.unesp.br/handle/11449/151310?locale-attribute=en] (accessed on 1 March 2023).

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
