# Peer review of "Staphylococcus aureus and CA-MRSA Carriage among Brazilian Indians Living in Peri-Urban Areas and Remote Communities"

_antibiotics, 2023, doi:10.3390/antibiotics12050862_

Round 1

Reviewer 1 Report

I appreciate the Authors performing this study, which involves a number of indigenous groups in Brazil. As the Authors mention, individuals in this population are often frequently overlooked in scientific and health endeavors and suffer from health inequity and poor health outcomes overall. 

Major Comments:

1. The Authors conclude that a number of factors possibly associated with certain ethnic groups but not others - such as habits and customs, poor housing conditions, hygiene, sanitation, etc - may be associated with Staph aureus carriage and possible spread.  But they don't really define these variables in the "Methods" section. What is meant by "Brickhouse", "sewerage system", "Distance from a health unit", etc. Many readers won't know what urucum and jenipapo are.  What is meant by "number of daily baths" - are people really taking 3-4 baths a day or is this hand washing?  All variables analyzed should be clearly defined in the "Methods" section. 

2. Please avoid the use of pejorative words such as "peculiar" (line 194 and 298) to describe the customs of indigenous persons. 

Minor Comments:

1. Line 35: MRSA infections are harder to treat that which other infections? Please explain.

2. Line 37: I believe you mean "sensitive", not "sensible".

3. Line 41 - 42: CA-MRSA outbreaks were first described among these communities... Please provide a bit more explanation.

4. Line 50: Please be careful of your wording here.  I would not use "indigenous" and "native" interchangeably.  Please stick to one term. 

5. Table A1.

- The Authors use both commas and periods to indicate decimal places.  Please pick one.

- Under "Predictors" and "Univariate Analysis", "Compl" should be "Complete".

- Should be "Habits and Customs", not "Habits and Costums".

6. Line 209: You mean "21" rather than "2 1 clusters".

Author Response

April 21, 2023.

The authors thank Reviewer 1 for the careful reading of the manuscript and valuable suggestions that helped improve the manuscript. The changes suggested by the reviewer and some updates are highlighted in yellow in the text. We address each comment separately below.

Reviewer 1 – round

I appreciate the Authors performing this study, which involves a number of indigenous groups in Brazil. As the Authors mention, individuals in this population are often frequently overlooked in scientific and health endeavors and suffer from health inequity and poor health outcomes overall.

Major Comments

  1. The Authors conclude that a number of factors possibly associated with certain ethnic groups but not others - such as habits and customs, poor housing conditions, hygiene, sanitation, etc - may be associated with Staph aureus carriage and possible spread. But they don't really define these variables in the "Methods" section. What is meant by "Brickhouse", "sewerage system", "Distance from a health unit", etc. Many readers won't know what urucum and jenipapo are.  What is meant by "number of daily baths" - are people really taking 3-4 baths a day or is this hand washing?  All variables analyzed should be clearly defined in the "Methods" section.

We thank the reviewer for the comment. All variables analyzed are included in the Methods section. Regarding the question about the number of daily baths, indigenous people living in remote areas have the habit of bathing in the river and it is common that they take many baths throughout the day. So, we refer specifically to the frequency of baths.

  1. Please avoid the use of pejorative words such as "peculiar" (line 194 and 298) to describe the customs of indigenous persons.

We apologize for the mistake. This word was changed to specific.

Minor Comments

  1. Line 35: MRSA infections are harder to treat that which other infections? Please explain.

Methicillin-resistant Staphylococcus aureus (MRSA) are a cause of staphylococcal infections that are difficult to treat because of resistance to some antibiotics.

  1. Line 37: I believe you mean "sensitive", not "sensible".

We apologize for the mistake. This word was corrected.

  1. Line 41 - 42: CA-MRSA outbreaks were first described among these communities... Please provide a bit more explanation.

Outbreaks of CA-MRSA were first described among indigenous communities. Important evidence occurred in the late 1980s, when a strain of MRSA arose spontaneously in remote Aboriginal communities of Western Australia. Additionally, the incidence of CA-MRSA has been high among indigenous populations living in remote areas of different countries.

  1. Line 50: Please be careful of your wording here. I would not use "indigenous" and "native" interchangeably. Please stick to one term.

We made the suggested correction, opting for the use of the word Indians.

  1. Table A1.

- The Authors use both commas and periods to indicate decimal places.  Please pick one.

- Under "Predictors" and "Univariate Analysis", "Compl" should be "Complete".

- Should be "Habits and Customs", not "Habits and Costums".

We made all suggested adjustments; we chose to use periods to indicate decimal places.

  1. Line 209: You mean "21" rather than "2 1 clusters".

The sentence was corrected.

Reviewer 2 Report

Minor Comments

In this research article, the authors investigate the prevalence of colonization with S. aureus and CA-MRSA among Brazilian Indians.  I would recommend the following comments to the authors. The manuscript is well-written, and the experimental design/data analysis is robust.  I would recommend the following comments to the authors.

Point 1: It would be better to rewrite sentences 27-30. Twenty-one clusters …… prevalence of S. aureus in this population.

Point 2: It would be nice to draw a detailed workflow chart. Otherwise, it would be difficult for the reader to capture the overall picture of the study. Overall, I could not really fault the experiments or the interpretation.

Good Luck

Author Response

April 21, 2023.

The authors thank Reviewer 2 for the careful reading of the manuscript and valuable suggestions that helped improve the manuscript. The changes suggested by the reviewer and some updates are highlighted in green in the text. We address each comment separately below.

Reviewer 2

Major Comments

In this research article, the authors investigate the prevalence of colonization with S. aureus and CA-MRSA among Brazilian Indians.  I would recommend the following comments to the authors. The manuscript is well-written, and the experimental design/data analysis is robust.  I would recommend the following comments to the authors.

  1. It would be better to rewrite sentences 27-30. Twenty-one clusters …… prevalence of S. aureus in this population

We thank the reviewer for the comment. The sentence was rewritten: “PFGE analysis identified 21 clusters among the S. aureus isolates and MLST analysis showed a predominance of sequence type 5 among these isolates. Our study revealed a higher prevalence of S. aureus carriage among Indians of the Shanenawa ethnicity (41.1%). Therefore, ethnicity appears to be associated with the prevalence of S. aureus in these populations.”

  1. It would be nice to draw a detailed workflow chart. Otherwise, it would be difficult for the reader to capture the overall picture of the study. Overall, I could not really fault the experiments or the interpretation.

We thank the reviewer for the suggestion. A workflow of the methodological steps was included in the article.

Figure S1.  Workflow of the methodological steps.

Reviewer 3 Report

In this manuscript, authors investigated the emergence of community-acquired methicillin-resistant Staphylococcus aureus (CA-MRSA) infections among indigenous populations living in extreme poverty. The need for healthcare resources further aggravates the situation. This study aimed to investigate the prevalence of S. aureus and CA-MRSA colonization among Brazilian Indians, which could help devise preventive measures for such infections. The study found that out of the 931 specimens (nasal and oral) screened from 400 indigenous individuals, S. aureus was cultured in 190 isolates (47.6%). Furthermore, CA-MRSA was found in three isolates (0.7%), all SCCmec type-IV, from indigenous individuals in remote hamlets. The study also identified 21 clusters among the S. aureus isolates, predominating isolates presenting Sequence-Type 5. The prevalence of S. aureus colonization was higher in Shanenawa ethnicity individuals. The study's findings emphasize the need for active surveillance and preventive measures to control the spread of CA-MRSA infections among indigenous populations. It also highlights the importance of healthcare resources and policies that address healthcare disparities and improves healthcare access for marginalized people. The manuscript is well-written, and the experiments are well-designed and well-executed. However, some areas could be improved to enhance the quality and impact of the manuscript.

Abstract:

1.              Page 1. Line 17, For CA-MRSA, authors should use the extended form when using an acronym for the first time.

Introduction:

1.              The introduction should highlight the research gap in the current literature., e.g., the need to investigate CA-MRSA infections among indigenous populations living in extreme poverty.

Results:

1.              Strain name should be written in Italics throughout the manuscript (e.g., Page 2. Line 88.).

2.              Data from Table A1 (Poisson regression model for the analysis of risk factors associated with S. aureus) under the section heading clinical variable showed 100 percent similarity from other sources that confer this data might be copied from a published source.

Other:

1.              Similarity index is very high, even more than 17%; authors should reduce similarity from the manuscript.

Author Response

April 21, 2023.

The authors thank Reviewer 3 for the careful reading of the manuscript and valuable suggestions that helped improve the manuscript. The changes suggested by the reviewer and some updates are highlighted in pink in the text. We address each comment separately below.

Reviewer 3 – round

In this manuscript, authors investigated the emergence of community-acquired methicillin-resistant Staphylococcus aureus (CA-MRSA) infections among indigenous populations living in extreme poverty. The need for healthcare resources further aggravates the situation. This study aimed to investigate the prevalence of S. aureus and CA-MRSA colonization among Brazilian Indians, which could help devise preventive measures for such infections. The study found that out of the 931 specimens (nasal and oral) screened from 400 indigenous individuals, S. aureus was cultured in 190 isolates (47.6%). Furthermore, CA-MRSA was found in three isolates (0.7%), all SCCmec type-IV, from indigenous individuals in remote hamlets. The study also identified 21 clusters among the S. aureus isolates, predominating isolates presenting Sequence-Type 5. The prevalence of S. aureus colonization was higher in Shanenawa ethnicity individuals. The study's findings emphasize the need for active surveillance and preventive measures to control the spread of CA-MRSA infections among indigenous populations. It also highlights the importance of healthcare resources and policies that address healthcare disparities and improves healthcare access for marginalized people. The manuscript is well-written, and the experiments are well-designed and well-executed. However, some areas could be improved to enhance the quality and impact of the manuscript.

Comments

Abstract

  1. Strain name should be written in Italics throughout the manuscript (e.g., Page 2. Line 88.).

We apologize for the mistake. This was corrected.

Results

  1. Data from Table A1 (Poisson regression model for the analysis of risk factors associated with S. aureus) under the section heading clinical variable showed 100 percent similarity from other sources that confer this data might be copied from a published source.

Dear reviewer, the document showing 100 percent similarity with the results reported in this article refers to the author’s complete doctoral thesis. At public universities in Brazil, all theses and master’s and doctoral dissertations are stored in the university’s digital database.

Others

  1. Similarity index is very high, even more than 17%; authors should reduce similarity from the manuscript.

Dear reviewer, the data of this article are original and have not been published in scientific journals. The only document that contains these data is the doctoral thesis of Lígia Maria Abraão, which must be kept in the thesis database of the university.

Round 2

Reviewer 1 Report

I thank the Authors for their diligent revision.